# Immunomodulatory Effects of Spherical Date Seed Pills Industrially Fabricated on RAW264.7 Cells

**DOI:** 10.3390/foods12040784

**Published:** 2023-02-12

**Authors:** Ibrahim Khalifa, Fahad K. Aldhafiri

**Affiliations:** 1Food Technology Department, Faculty of Agriculture, Benha University, Moshtohor 13736, Egypt; 2Department of Public Health, College of Applied Medical Sciences, Majmaah University, Majmaah 15341, Saudi Arabia

**Keywords:** immunomodulation, byproducts, date seed pills, large-scale, RAW264.7 cells

## Abstract

Dates have been demonstrated to display a variety of bioactivities and are rich in polyphenols. In this work, we assessed the underlying immunomodulatory effects of date seed polyphenol extracts that had been industrially encapsulated and fabricated into commercial pills in RAW264.7 macrophages using the NF-κB and Nrf2 signaling pathways. The outcomes showed that in RAW264.7 cells, the date seed pills effectively stimulated nuclear translocation of NF-E2–related factor 2 (Nrf2) and NF-κB, along with downstream cytokines (IL-1β, TNF-α, IL-6, and IFN-γ), ROS ratios, and SOD activity. It is interesting to note that the encapsulated pills activated Nrf2 nuclear translocation more effectively than the non-encapsulated ones did. Additionally, pills at 50 µg mL^−1^ improved immunological responses, but pills at 1000 µg mL^−1^ prevented macrophages from becoming inflamed. These results showed that the immunomodulatory effects were differently impacted by commercial date seed pills, a finding which was related to the large-scale manufacturing of the pills and the incubation concentrations used. These results also shed light on a new trend of using food byproducts as an innovative supplement.

## 1. Introduction

In addition to various regions, namely Europe (the southern part), America (the central and southern part); date fruits, or *Phoenix dactylifera* L., are significant cultivated products in North Africa and the Mediterranean region. Date fruits have long been regarded as the perfect meal supplement by Middle Easterners because of their high sugars, dietary fiber, macronutrients, and micronutrient levels [1]. Date fruits’ maturity and ripeness stage affect their sweetness and texture. The three developmental stages of date fruit—aged firm (Besser or Khalal), semi-ripened (Rutab), and ripened (Tamr)—are typically harvested and marketed [2]. In addition to their nutritive value, date fruits have been linked to numerous bioactive benefits, including antioxidant, antimutagenic, anti-inflammatory, anti-cancer, anti-microbial, and immunostimulant effects [3]. Numerous polyphenols found in date fruits, including phenolic acids, hydroxycinnamates, proanthocyanidin oligomers, and flavonoid glycosides, are thought to be responsible for the fruit’s health benefits [4]. Recent research has shown that several polyphenols found in date fruit, such as cinnamic acid derivatives and pelargonin, boost the cellular resistant assay [5].

Owing to its extra nutritional content, date fruits are an essential dessert that are marketed as a variety of foods. However, around 12–15% of the date fruits are seeds, which are mostly considered as the fruits’ byproduct and have not been economically valued yet. The region’s Bedouin tribes used to drink infusions made from roasted date seeds, but they have otherwise not been significantly used by humans to date. According to earlier research, date seeds have a high polyphenolic content and exceptional antioxidant activity. Numerous animal studies document biological benefits, including a reduction in oxidative damage, an improvement in memory and comprehension in Alzheimer’s models, and an anti-diabetic effect [6,7]. More in-depth research on the biochemical makeup of dates is of interest given their nutritional importance and the growing knowledge regarding the positive health impacts of uncooked fruits.

According to epidemiological research, the polyphenols in diets and supplements made from fruit may lower the hazards of cancer, diabetes, and cardiovascular illnesses [8]. It has been shown that polyphenols control inflammation via a variety of methods, including oxidative stress reduction through the scavenging of free radicals, regulation of inflammatory intermediaries like cytokines, and modification of signing pathways that have elaborate consequences on fruit tenderness [9]. Studies on immunity have received more attention in recent decades, and various signaling pathways that are involved in immunoregulation by macrophages have been examined [10]. Nuclear factor kappa-B (NF-*κ*B) regulates several examples of immune-activating cytokines, including tumor necrosis factor-*α* (TNF-*α*), interferon-*γ* (IFN-*γ*), interleukin-6 (IL-6), and interleukin-1*β* (IL-1*β*); it is an oxidative strain-receptive transcription factor [11]. Chronic inflammatory diseases may cause NF-*κ*B dysregulation, and drugs that control NF-*κ*B may prove to be innovative treatment options. Furthermore, another crucial controller of cellular oxidative anxiety called NF-E2-related factor-2 (Nrf2) is a catalyst for cellular immune reactions [12]. It has been demonstrated that polyphenols and their derivatives have the potential to act as immunoregulatory agents. In RAW 264.7 cells, hydroxycinnamic acids can prevent the production of pro-inflammatory cytokines, such as TNF-*α*, IL-6, and IL-1*β*, as well as mitigate the translocation of NF-*κ*B into the nucleus [13]. Kim et al. discovered that neochlorogenic acid was capable of mitigating the pro-inflammatory effects and phosphorylation of NF-*κ*B p65 in BV2 microglial cells [14]. Additionally, both in vitro an in vivo trials have revealed that catechin drastically decreases the buildup of intracellular reactive oxygen species (ROS) and guards against oxidative injury to the stomach caused by ketoprofen [15].

These compounds are unstable, though. In this situation, microencapsulation technologies could help stabilize polyphenols for use in industrial applications. The process of microencapsulation is used to turn liquid solutions into powders for simpler handling and to increase the stability of bioactive compounds by shielding them from oxygen, water, light, and other environmental factors [16]. Therefore, we proposed the possibility that date polyphenol pills may modulate the immune system by controlling the NF-*κ*B and Nrf2 signaling pathways. In this study, we investigated and assessed the various immunomodulatory effects of industrially produced date pills and identified a putative immunoregulatory mechanism involving Nrf2 nuclear translocation and NF-*κ*B on RAW264.7 macrophages. The findings of this study shed light on how date polyphenols behave functionally.

## 2. Materials and Methods

### 2.1. Chemicals and Kits

PVDF layers (Immobilon-P, 0.45 μm), dimethyl sulfoxide (DMSO), and 3-(4,5-dimethylthiazol-2-yl)-2,5-diphenyltetrazolium bromide (MTT) were acquired from Sigma-Aldrich (Saint Louis, MO, USA). The following items were acquired from Thermo Fisher Scientific: Penicillin-streptomycin mix, Dulbecco’s altered Eagle’s medium (DMEM), trypsin (0.25% EDTA), fetal bovine serum (FBS), tris-buffered saline (TBS), and phosphate-buffered saline (PBS) (Shanghai, China). Proteintech Group, Inc. sold anti-NF-κB p65, anti-lamin B1, and anti-Nrf2 antibodies for rabbits. Bioscience Technology provided sodium lauryl sulphate, tris base, acrylamide, glycine, ammonium persulfate, bromophenol blue, and N,N’-methylene acrylamide (Shanghai, China). Proteintech provided TNF-α, IFN-γ, IL-6, and IL-1β enzyme-allied immunosorbent test (ELISA) implements (Wuhan, China). Date seeds of the Siwa variety were acquired from the Shalli Company (Siwa, Marsa Matruh, Egypt) in March 2022; the seeds were harvested at room temperature 12 h prior to arriving at our lab where they were then prepared immediately without being stored. All the other substances employed in this investigation, including the solvents, were of analytical grade.

### 2.2. Commercial Fabrication of Date Pills Rich in Polyphenols

The date seeds were washed (Rotary Washing and Soaking Machine, Markoom, Turkey), dried (OT-2F, Gilson Company Inc, Columbus, OH, USA), roasted (Payo Roasting Machine, Dabiige Co., Davao, Philippines), milled (3 HP Commercial Grinder, Alishan Co., Delhi, India), seized to 45 mesh for more homogenization and uniformity, and formulated to prepare the end formula of the pills. After that, the pills were formed using a ZP-15E-19E automatic rotary tablet press machine (China) using the production line of Tadawina Company (Shubra, Egypt). Gum arabic and maltodextrin were used to fabricate the pills, with each pill containing a ratio of 400, 500, and 100 mg per g of polyphenols, maltodextrin, and gum arabic, respectively. Figure 1 summarizes the overall steps required to produce date pills that are rich in polyphenols using a pressing technique and a blend of gum arabic and maltodextrin as carriers.

### 2.3. Analysis of the Polyphenols of Date Pill Extracts

The polyphenols of the date seed pills were extracted using our previous method [17]. Chromatographic analysis was conducted on a Waters Acquity UPLC-PDA system (Waters, Milford, CT, USA) with the aid of an Acquity BEH C18 column (10 × 1 mm i.d., 1.7 µm). Phase A (H_2_O- HCOOH, 99:1, *v*/*v*) and phase B (MeOH- HCOOH, 99:1, *v*/*v*) made up the mobile phase (solvent B). The stream ratio was 0.08 mL per min, and the oven was kept at 35 °C. The elution schedule was as follows: isocratic for 1 min with 2% B, 2–18% B (1–7 min), isocratic with 18% B (7–9 min), 18–30% B (9–12 min), isocratic for 2 min with 30% B, 30–75% B (14–27 min), and then cleaning and reconditioning of the column. The infusion had a 2 µL volume. An electrospray basis and an ion trap mass analyzer were included in the Bruker Daltonics Amazon (Bruker, Germany) mass spectrometer that was used for the ESI-MS/MS analysis and connected to the UPLC-PDA system. A negative ion manner (capillary voltage of 4.5 kV; end plate offset of 500 V; temperature of 200 °C; nebulizer gas of 10 psi, and dry gas of 5 µL min^−1^) and positive ion mode (capillary voltage of 2.5 kV; end plate offset of 500 V; temperature of 200 °C; nebulizer gas of 10 psi, and dry gas of 5 µL min^−1^) were used to (perform the ESI-MS/MS analysis) the MS2 experiments’ collision energy for fragmentation, which was adjusted to 1. Retention periods, UV-visible spectra, and MS spectra were used to identify the samples. We compared our data with polyphenols-based data found using standard components that were identified in related references [18,19,20]. The concentrations were estimated using the area of each peak at 280, 320, and/or 360 for each of protocatechuic acid and monomers and oligomers of flavan-3-ol units liberated via phloroglucinolysis, hydroxycinnamic acids, and flavones and flavonols, respectively. This was accomplished with the aid of exterior standardization curves that were obtained using a comparable method or analogous substances, namely, quercetin 3-glucoside for flavones and flavonols and caffeic acid for hydroxycinnamic derivatives (0.04-50 g Kg^−1^). Phloroglucinolysis retains the stereochemistry and potential substituents of proanthocyanidin terminal units while releasing extension units as the matching phloroglucinol derivatives. Total flavan-3-ol content was computed as the sum of constitutive units liberated during phloroglucinolysis. The data sets examined were collected, aligned, and normalized via MarkerLynx software. Peak width at 5% height of 1 s, peak-to-peak baseline noise of 1, noise elimination of 6, and intensity threshold of 10,000 were utilized to capture metabolite peaks. A retention time window of 0.2 min and a mass window of 0.05 Da were utilized to align the data.

### 2.4. Cell Culture

The Chinese Academy of Sciences’ Cell Bank of Form Culture Compilation sold the RAW264.7 murine macrophage cell culture and an Abelson leukemia viral-stimulated polyp cell culture (Shanghai, China). Cells were hatched in a 37 °C hatchery with CO_2_ (5%) and cultivated in DMEM accompanied with FBS (10%, *v*/*v*), penicillin (100 U mL^−1^), and streptomycin 100 (μg mL^−1^). The media were changed three times a week and were planted at a previously mentioned thickness (roughly 1.0 × 10^5^ cells per cm^2^). Before usage, all cells underwent more than three passages.

### 2.5. Cell Viability Analysis

An MTT test was utilized to measure cell capability. At an intensity of 1 × 10^5^ cells mL^−1^, RAW264.7 cells were sown in 96-well bowls and hatched for 12 h at 37 °C. The cells were later re-incubated in serum-free media that had been supplemented with pill extracts at doses of 50, 250, 500, 750, and 1000 µg mL^−1^ after the supernatants had been aspirated. The blank control was medium alone. Following a 24-h maturation period, the upper phase was removed, each well was mixed with 100 µL of MTT solution, and all of the cells were incubated at 37 °C for 4 h. A 100 µL solution of DMSO (100%) was subsequently used to lyse each cell at room temperature (25 °C) for 10 min. Utilizing a Varioskan Flash microplate reader, the optical densities of the control cell (Ac) and each of the pill-based cell wells (At) were determined at 570 nm (Thermo Fisher Scientific, Waltham, MA, USA). For all studies, six separate tests were run. As previously mentioned, the equivalence Cv (%) = (At − Ac)/Ac × 100 was used to compute cell viability (Cv) [11].

### 2.6. Cytokine Analysis

Using commercial ELISA kits, the ratios of IL-1*β*, IL-6, IFN-*γ*, and TNF-*α* in the matrix were evaluated. Briefly, the RAW264.7 cells were preserved for 24 h in a 5% CO_2_ hatchery at 37 °C after being pre-hatched in a 12-well plate at an intensity of 1.0 × 10^6^ cells per well. Culture media were collected and transferred into 96-well plates covered with 100 μL of each of the antibody blends for IL-1*β*, IFN-*γ*, TNF-*α*, and IL-6 and held at 37 °C for 60 min. After four washes, 100 μL of horseradish peroxidase (HRP) blend was inserted, and the bowls were then hatched for 40 min at 37 °C. The bowls were then dark-hatched at 37 °C for 15 min before 100 μL of 3,3′,5,5′-tetramethylbenzidine (TMB) substratum mix was inserted. Finally, five minutes after adding a stop solution, the transmission density at 450 nm was detected using a Varioskan Flash microplate reader (Thermo Fisher Scientific, Waltham, MA, USA). Pill extract-free incubation was used for the control group. Standard curves were used to calculate the cytokine levels. There were three duplicates of each test run. The concentrations of cytokine were presented as picograms each mL (pg mL^−1^) of sample.

### 2.7. Intracellular ROS Analysis

ROS generation was evaluated using a modified version of the previously published 2,7-dichlorofluorescein diacetate (DCHF-DA) test [21]. The amount of intracellular ROS may be measured by observing the fluorescence of DCF, which is produced when nonfluorescent DCFH is oxidized to become fluorescent DCF. Briefly, RAW264.7 cells were planted in 24-well bowls at an intensity of 1.0 × 10^5^ cells per well. For 20 min, the cells were subjected to 50, 250, 500, 750, and 1000 μg mL^−1^ of pill extract. For the pill-free sample, a medium without polyphenols was employed. The media were then evenly mixed with a fluorescent probe (10 µM), and each cell was hatched at 37 °C for an extra 20 min.

To eliminate any DCFH-DA that had not penetrated the cells, the cells were then rinsed three times with warm PBS. Using the FACS Aria II flow cytometer, the DCF fluorescence of the cell suspension was quantified (Becton Dickinson, Franklin Lakes, NJ, USA). The DCF fluorescence strength represents the quantity of intracellular ROS. The trials were conducted three times and the outcomes are shown as mean estimates ± SD.

### 2.8. Superoxide Dismutase (SOD) Analysis

According to the instructions provided with the kit, SOD rates were measured using ELISA (KeyGen Biotech Co., Ltd., China). In a 96-well plate, 20 μL of pills and 200 μL of employed blend were combined uniformly. Then, the enzymatic blend (20 μL) was inserted, and the pill-based samples were hatched at 37 °C for 20 min. As the plain control, 20 μL of ddH_2_O was utilized in place of the 20 μL sample, and as the test control, 20 μL of thinned buffer was utilized in place of the 20 μL enzyme operating blend. At a wavelength of 450 nm, the measurements were performed using an ELISA microplate reader. Each sample’s SOD action was calculated. One unit of SOD activity was defined as the enzyme quantity necessary to block 50% of the lessening interaction between superoxide anions and H_2_O-soluble tetrazolium salt-1 (WST-1) in a sample blend (20 μL).

### 2.9. Western Blot Assessment

RAW264.7 cells were pre-hatched in a 6-well bowl for 4 h in a 5% CO_2_ hatchery at an intensity of 1.0 × 10^6^ cells per well. The cells were then preserved for 24 h in media that had been supplemented with 50, 250, 500, 750, and 1000 μg mL^−1^ of pill extract. After that, the cells were collected, and a nuclear and cytoplasmic extraction kit was used to make nuclear extracts (SC-003, Inventbiotech, Shanghai, China). The supernatants were collected for further protein analysis after the isolates were cold-centrifugated at 4 °C for 20 min at 11,000× *g*. A 10% DYY-6C SDS-PAGE (Keyichuang Biotechnology, Guiyang, China) at 80 V for 2 h was utilized to divide the nuclear protein. After that, 12.5 and 7.5 μL of nucleoprotein extract and PageRuler Prestained Protein Ladder, respectively, were added to the gel. A PVDF layer was employed to transport the proteins at 350 mA for 1.5 h with the aid of a BIO-RAD Mini-Trans-Blot Cell (Bio-Rad, Hercules, CA, USA). The PVDF layer was hatched with initial antibodies disbanded in 0.5% skim milk blend at 4 °C for an extra 2 h after being clogged with 5% skim milk in TBS covering 0.5% Tween 20 (TBS-T). The anti-Nrf2, anti-NF-κB p65, and anti-lamin B1 primary antibodies were thinned at a rate of 1:500. The PVDF membrane was cleaned with TBS-T solution and then hatched for 1 h at RT with a secondary goat anti-mouse IgG antibody that was HRP-labeled. The PVDF layer was then cleaned with TBS-T for 20 min. After being submerged in an enhanced chemiluminescence (ECL) fluid that had been freshly made before usage, the membrane was then left to sit at RT for 3 min before being imaged using a LAS-4000 MINI biomolecule imager (Fuji, Japan). The band intensity in relation to the internal reference lamin B1 represents the protein expression. Three tests were conducted on each sample, and the findings were portrayed as means ± SD.

### 2.10. Statistical Analysis

The findings were shown as means ± SD. Using SPSS 21.0 Statistics, a one-way analysis of variance (ANOVA) followed by post hoc Tukey’s checks for numerous differences was utilized to evaluate the data (SPSS Inc., Chicago, IL, USA). A result was deemed statistically significant if it had a *p-*value of < 0.05.

## 3. Results and Discussion

### 3.1. Quantification of Date Seed Polyphenols after Encapsulation into Spherical Pills

The peak regions of each polyphenolic component were used to determine its concentration. Prior to depolymerization, the date seeds contained 1.868 ± 0.03 g kg^−1^ of individual polyphenols (Table 1). The concentration of protocatechuic acid was 0.082 ± 0.003 g kg^−1^; the concentration of flavan-3-ol monomers and oligomers was 1.68 ± 0.03 g kg^−1^ (represented as epicatechin equivalents); the concentration of flavones and flavonols was 0.045 ± 0.001 g kg^−1^ (represented as quercetin 3-glucoside equivalents); and the concentration of hydroxycinnamic acids was 0.06 ± 0.001 g kg^−1^ (represented as caffeic acid equivalents). The components that were the most prevalent were flavan-3-ols. With a typical level of polymerization, the total quantity of flavan-3-ols determined after depolymerization was 49.3 ± 1.8 g kg^−1^. This included 45.9 ± 1.1 g kg^−1^ of epicatechin and 3.4 ± 0.4 g kg^−1^ of catechin (Table 1). Consequently, the total quantity of flavonoids and phenolic substances in the date seeds was about 51.168 g kg^−1^. Our results are well matched with those of a previous study [18].

### 3.2. Cell Viability and Cytokine Expression

To examine how the pills affected the viability of RAW264.7 macrophage cells, MTT tests were used (Figure 2). The findings showed that the pill extract samples were noncytotoxic at doses of 50–1000 μg mL^−1^. Additionally, incubation with the pills increased the viability of the cells by 80, 81, 86, 99, and 108% for 50, 250, 500, 750, and 1000 μg mL^−1^, respectively. These outcomes are consistent with earlier research which demonstrated that treating the same cell line (RAW264.7) with 0.25–0.4 mg mL^−1^ of purslane extract increased, rather than decreased, cell viability [22].

Guo*,* et al. [23] also found that five (50–1000 μg mL^−1^) different concentrations of thinned peach cured by several dehydration approaches significantly increased cell viability. Our results showed that the corresponding Maillard adducts in the date seed samples after roasting (160 °C/15 min) were not cytotoxic. The impacts of low and high doses of date seed pills on the macrophage immune response were clarified in subsequent tests using the same doses of 50–1000 μg mL^−1^ pill extract polyphenols. The innate immune system’s fight against infections is greatly aided by the cytokines released by macrophages [24]. In the current study, cytokine production (TNF-*α*, IL-1*β*, IFN-*γ*, and IL-6) in various pill-incubated RAW264.7 cells was assessed using ELISA kits. According to Figure 3A, cultivation with 50 µg mL^−1^ of pills boosted IFN-*γ*, TNF-*α*, and IL-1*β* production by 35, 33.3, and 38.8%, respectively. This finding is in line with an earlier study that found that docosahexaenoic acid triggers the manifestation of immune cytokines in RAW264.7 cells at concentrations between 1.2 and 3.0 μM [11].

Because the levels of catechin, epicatechin, and protocatechuic acid in the date seed pills (Table 1) were similar and significantly high, it is noteworthy that the pills were comparable to thinned peach in their ability to increase cytokine excretion [23]. Catechins have been shown to affect immunity by controlling the levels of IL-10 and-1*β*, alongside TNF-*α* and RIG-I cytokine nodding [25]. Our results support earlier studies which show that these polyphenols can have strong anti-inflammatory effects by altering cytokine production in a range of cell lines, including animal models [14]. Notably, the Maillard reaction adducts may have striking impacts on polyphenol stabilization since 50 µg mL^−1^ pill incubation caused the smallest increase in production of these cytokines. Intriguingly, the accumulation of TNF-*α*, IFN-*γ*, and IL-1*β* in 1000 µg mL^−1^ pill-incubated cells (RAW264.7) was decreased by 18.18, 116.6, and 175.2%, respectively, with incubation with the pills producing a dose-dependent effect. An excessive level of polyphenols may interact with bioactive proteases, causing them to dissolve and undergo functional modifications that have an inhibitory effect on cytokine production [26]. We used molecular docking using MOE based on our previous method to support this hypothesis [27,28]. Figure 3B portrays the interaction between epicatechin (the key polyphenol of the date seeds pills) and proteases. In brief, epicatechin interacted with the proteases by three H-bonds and π–π stacking. This outcome is consistent with a prior study which demonstrated that purslane extracts at comparatively superior doses of 0.4 mg mL^−1^ dramatically restrained the production of TNF-*α* and IL-6 [22]. Fluctuations in the level of the cytokine IL-6 (Figure 3A) were strongly inhibited by pill incubation at both 50–1000 µg mL^−1^. In general, at healthy doses, IL-6 has little autocrine impact on immune cells and has no influence on the cell’s ability to release further cytokines. Nevertheless, it has been hypothesized that elevated IL-6 concentrations might cause irritation by regulating the release of other cytokines. In the current investigation, the date seed pills significantly lowered the level of IL-6 in cells (RAW264.7), reducing the cells’ defensive response. The catechins and epicatechins in the pills were mostly responsible for this action, a finding which might be confirmed by a prior discovery that the green tea catechin epigallocatechin gallate prevented IL-6 release in HEK293T cells [8].

### 3.3. ROS Accumulation and SOD Activity

The primary oxidation products in cells are ROS, which are essential actors in cell signaling events. Their buildup triggers macrophages to endure extra apoptosis or autophagy, which increases the onset of cell death [9]. Figure 4 shows the impact of date seeds pills on the production of ROS in RAW264.7 cells. When exposed to 50 µg mL^−1^ pills, the formation of intracellular ROS was reduced by 8.61%. The pills inhibited ROS by 7.75% less than the polyphenols from thinned peach, which inhibited ROS by 16.36%. Because they contained high concentrations of catechin, epicatechin, and phenolic acids, 50 µg mL^−1^ pills were found to significantly reduce the production of ROS in this study. According to previous studies, the main ingredients of *Moringa oleifera* leaf extracts (0.1 mg mL^−1^) that significantly reduced the formation of ROS in HEK293 cells were cryptochlorogenic acid and isoquercetin [29]. However, incubation with 100 µg mL^−1^ pills led to a notable increase by 61.5% in intracellular ROS generation, in which polyphenols at high concentrations behaved as pro-oxidants [30]. According to research by Robaszkiewicz et al. [31], flavonoids, which were also present in our pills, showed antioxidant benefits at small doses but increased oxidation at high doses (>50 μM) in A549 cells. According to the findings of Carvalho et al. [32], Maillard reaction adducts may function as pro-oxidants in the cell signaling response of RAW264.7 and offset the undue antioxidant impact of polyphenols.

Antioxidant enzymes like SOD are linked to the preservation of cellular redox equilibrium. Increasing the activity of antioxidant enzymes has been proven to lower ROS ratios in cells and thereby preserve intracellular redox equilibrium in several studies [33]. In this study, the influence of the cellular antioxidant enzyme SOD was examined to learn more about how the pills affect intracellular antioxidant levels. As shown in Figure 4, after being incubated with 50 µg mL^−1^ pills, intracellular SOD activity was increased by 7.8%, mostly due to the substantial amounts of epicatechin present in the date seed pills. The finding by Zhao et al. [34] that polyphenols from Kuding tea might aid in boosting the effects of SOD in the serum of mice corroborate this outcome. Nevertheless, incubation with 1000 µg mL^−1^ pills reduced SOD activity by 60.75%. The findings suggest that polyphenols might function as either antioxidants or pro-oxidants depending on the quantities of the polyphenol components. Redox stress may result from the pro-oxidant effects which can disrupt redox equilibrium [30].

### 3.4. NF-κB and Nrf2 Nuclear Translocation

Analyzing the nuclear translocation of p65 can help identify the impact of the NF-*κ*B pathway on macrophage biological activity. Therefore, to investigate how date seed pills affect the activation of the NF-*κ*B signaling pathway, nuclear NF-*κ*B p65 scores were measured via Western blotting. According to Figure 5, 50 µg mL^−1^ pill incubation initiated NF-*κ*B p65 nuclear translocation, which is similar to the discovery that 50 µg mL^−1^, 50 µg mL^−1^, and 2% of polysaccharides, ginsenosides, and fatty acids, respectively, separately upregulated nuclear NF-*κ*B manifestation to employ immunological activity in RAW264.7 cells [35]. Additionally, incubation with the pills boosted NF-*κ*B p65 nuclear translocation by 66%. Interestingly, nuclear NF-*κ*B p65 rates were decreased by 33% after 1000 µg mL^−1^ pill incubation. Nuclear NF-*κ*B p65 levels were significantly reduced by the pills. It may be inferred that high-dose pill incubation likely had pro-oxidant impacts at first, inducing inflammatory responses, and then acted as an anti-inflammatory agent by lowering the nuclear translocation of NF-*κ*B. Hence, incubating with higher doses of pills might prevent the onset and progression of the cellular inflammatory response in the presence of oxidative stress. An earlier study that found that high-dose (100 µg mL^−1^) phenolics from grape waste might reduce NF-*κ*B stimulation and mitigate lipopolysaccharide-induced oxidative strain in RAW264.7 cells provides support for this finding [36]. Additionally, sustained incubation with the high-concentration pills may cause I*κ*B to attach to p65 and enter the nucleus, thereby decreasing the affinity of p65 for associated genes and enabling it to separate from the binding site and enter the cytoplasm. As a result, NF-*κ*B p65 nuclear translocation was inhibited by incubation with high-dose pills.

In the intracellular environment, Nrf2 activation can improve the level of antioxidation and the maintenance of redox equilibrium. Western blotting was utilized to analyze the effects of date seed pills on nuclear Nrf2 levels and the findings are displayed in Figure 5. Cells cured with 50 µg mL^−1^ pills had higher nuclear Nrf2 levels, whereas cells cured with 1000 µg mL^−1^ pills had lower levels. In particular, incubation with 50 µg mL^−1^ pills increased Nrf2 nuclear translocation. Additionally, incubation with 1000 µg mL^−1^ pills reduced Nrf2 nuclear localization. According to Granato, Mocan and Câmara [30], high dietary polyphenol levels might have harmful pro-oxidative impacts and cause inflammation by activating the Nrf2 and NF-*κ*B signaling pathways. This is mostly caused by the elevated catechin concentration in the date seed pills. Numerous studies showing that catechin can prevent ketoprofen-stimulated oxidative injury to the gastric mucosa by modulating Nrf2 in vitro and in vivo might be used as evidence to support this finding.

Based on the aforementioned results, we hypothesized that the pills may potentially trigger cellular immunological responses by means of the NF-*κ*B and Nrf2 pathways (Figure 6). By activating I*κ*B kinase and boosting the phosphorylation of I*κ*B, which further increases ubiquitination under the control of ubiquitin ligase, incubation with date seed pills may influence the strength of the p65–I*κ*B*α* dimer in the cytoplasm of RAW264.7 cells. The p65 subunit is subsequently distributed into the nucleus where it attaches to point genes to improve transcription and have an impact on cytokine production. The following step involves ubiquitin ligase undergoing a conformational alteration and being identified and destroyed by a protease. Thus, cells react to the pills by activating NF-*κ*B through a classic route. An essential nuclear transcription factor and controller of the cellular redox balance is Nrf2. It has been demonstrated that a variety of signaling reactions may be induced by initiating the Kelch-like ECH-related protein 1 (Keap1)/Nrf2-ARE pathway. The inhibitory protein Keap1 is typically linked to Nrf2 in the cytosol under normal physiological circumstances. Pill incubation, on the other hand, stimulates the split of Nrf2 and Keap1, allowing Nrf2 to enter the nucleus and collaborate with the tiny Maf centrosome protein with ARE in developers, and ultimately influence the action of antioxidant enzymes and ROS buildup [37]. Additionally, the cells (RAW264.7) were hatched with 50–1000 µg mL^−1^ pills without experiencing any harmful effects. It was discovered that the pills exhibit bioactivity related to the activation and inhibition of NF-*κ*B and Nrf2 when incubated at the chosen doses. This bioactivity was strongly associated with the antioxidant and pro-oxidant actions of the pill models. According to de Camargo et al. [36], it has been hypothesized that colorimetrical antioxidant tests might have a tendency to verify NF-*κ*B stimulation tests, however our data do not support this theory. The bioactivity of a given pill is highly dependent on its dose, which was found to alter the oxidation–deoxidation balance of the cells and cause a range of cytokines and nucleic factors to react to stress in diverse ways.

In the meantime, the bioactive phenolic substances might penetrate the cells and immediately alter or initiate the nucleic factors, followed by the production of enzymes and cytokines. The question then becomes: is consuming a lot more polyphenols better or worse than consuming a lot less? The answer could be directly related to the unique redox environments present in the intracellular and extracellular environments. Therefore, based on our findings, we can assume that the pills may have advantages regarding the regulation of the redox status and inflammatory response that are related to an organism’s immune response.

## 4. Conclusions

According to the current investigation, encapsulation significantly enhanced the stability of the polyphenols present in date seed pills and had a variety of impacts on RAW264.7 macrophages. However, Nrf2 nuclear translocation was impacted by pills at 50 and 100 µg mL^−1^. Additionally, incubation with 50 µg mL^−1^ pills improved NF-*κ*B and Nrf2 nuclear translocation, encouraged the creation of the cytokines TNF-*α*, IFN-*γ*, and IL-1*β*, boosted intracellular SOD activity, and reduced ROS buildup. In contrast, NF-*κ*B and Nrf2 nuclear translocation was inhibited by incubation with 1000 µg mL^−1^ pills. Overall, it is possible to surmise that the pills controlled the NF-*κ*B and Nrf2 signaling pathways linked to the cellular stress response and redox homeostasis to mediate immunological activation. The findings showed that date seed pills had varying effects on the immunological responses in RAW264.7 macrophages. We are currently studying the adipocyte differentiation of our pills in 3T3-L1 cells. Moreover, we are planning to optimize the encapsulation of the date pills by reducing the carrier ratios and increasing the active ingredients. In addition, the binding between the carriers and active ingredients inside the pills ought to be elucidated, and a full characterization of the pills should also be considered.

## Figures and Tables

**Figure 1 foods-12-00784-f001:**
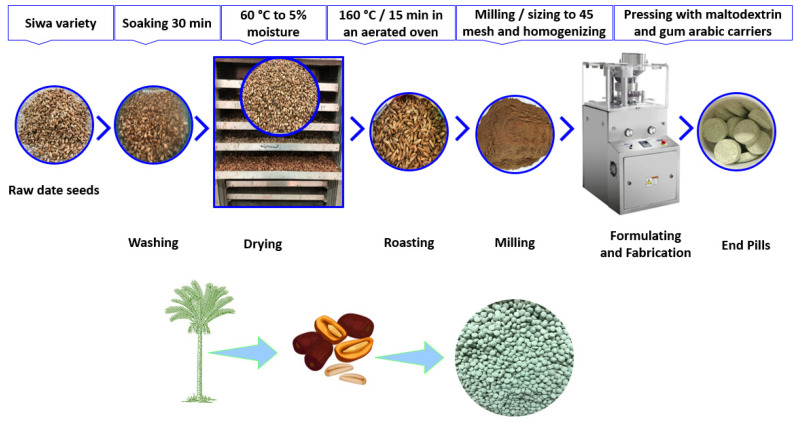
The overall steps required to produce date seed pills under large-scale conditions.

**Figure 2 foods-12-00784-f002:**
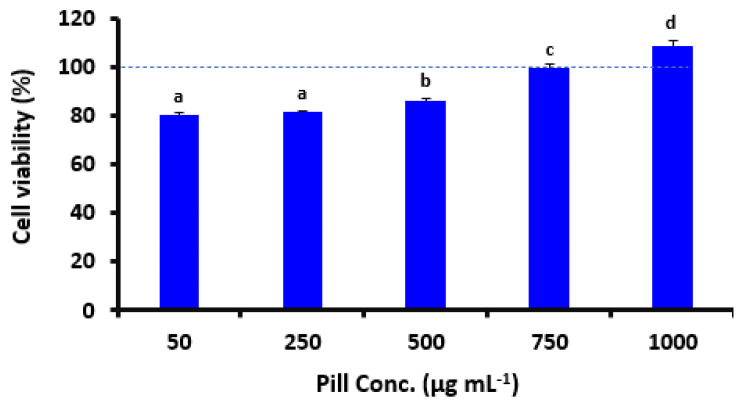
Effect of date seed pills at small and high doses. The polyphenol extracts of pills were extracted and used as a base of the concentration. The findings are presented as means ± SD. RAW264.7 cells were first planted at a dose of 1 × 10^6^ cells for each well in a whole medium for 24 h and later equipped with a medium comprising several doses of date seed pill polyphenols for an additional 24 h. The optical density (OD) estimates were defined after utilizing DMSO to soften each cell. Pill conc. refers to the concentration of the pill extracts used in our study, which were extracted after being formulated and formed. The same letters are insignificantly different (*p* > 0.05) following the Duncan’s multiple range tests.

**Figure 3 foods-12-00784-f003:**
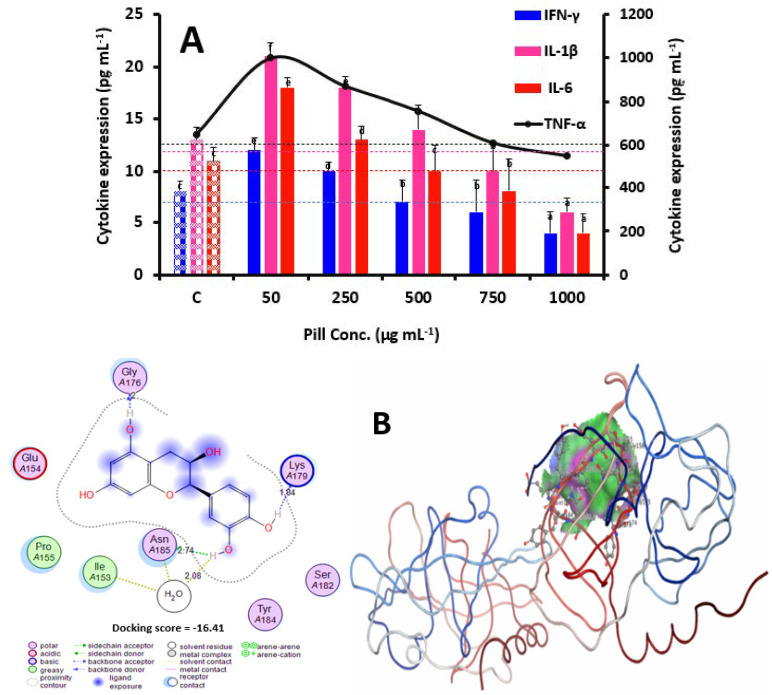
Effects of date seed pills on TNF-α, IL-1β, IFN-γ, and IL-6 cytokine production in RAW264.7 cells (**A**). Lamin B1 worked as an inner loading control. NF-*κ*B, nuclear factor kappa-B; Nrf2, NF-E2-linked factor-2. The findings are presented as means ± SD. Distinct letters reveal considerable variations at *p* < 0.05. The 2D- and 3D-interaction models between epicatechin (PubChem CID: 72276) and proteases (PDB: 2AS9) are also presented (**B**). Pill conc. refers to the concentration of the pill extracts used in our study, which were extracted after being formulated and formed. The same letters are insignificantly different (*p* > 0.05) following the Duncan’s multiple range tests.

**Figure 4 foods-12-00784-f004:**
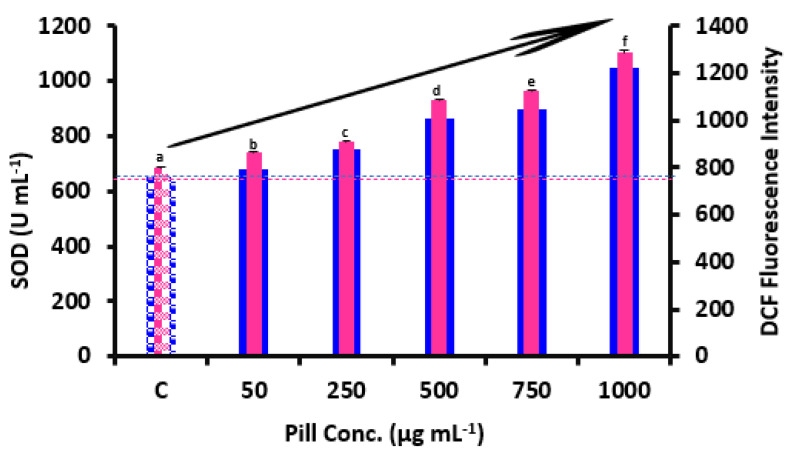
The impact of date seed pills on intracellular SOD action and ROS production in RAW264.7 macrophages. SOD, superoxide dismutase; ROS, reactive oxygen species. The findings are presented as means ± SD. Distinct letters reveal considerable changes at *p* < 0.05. Pill conc. refers to the concentration of the pill extracts used in our study, which were extracted after being formulated and formed. The same letters are insignificantly different (*p* > 0.05) following the Duncan’s multiple range tests.

**Figure 5 foods-12-00784-f005:**
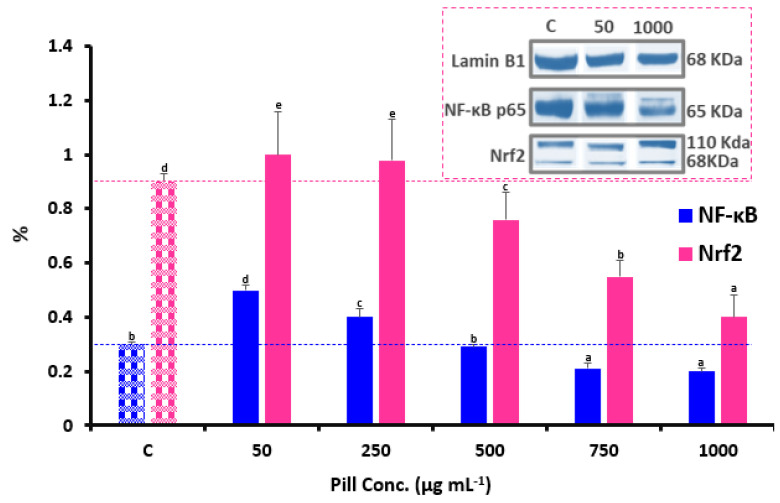
Effects of date seed pills on the relative protein intensity (%) of NF-κB p65 and Nrf2 and their Western blotting. The findings are presented as means ± SD. Distinct letters indicate significant differences at *p* < 0.05. Pill conc. refers to the concentration of the pill extracts used in our study, which were extracted after being formulated and formed. The same letters are insignificantly different (*p* > 0.05) following the Duncan’s multiple range tests.

**Figure 6 foods-12-00784-f006:**
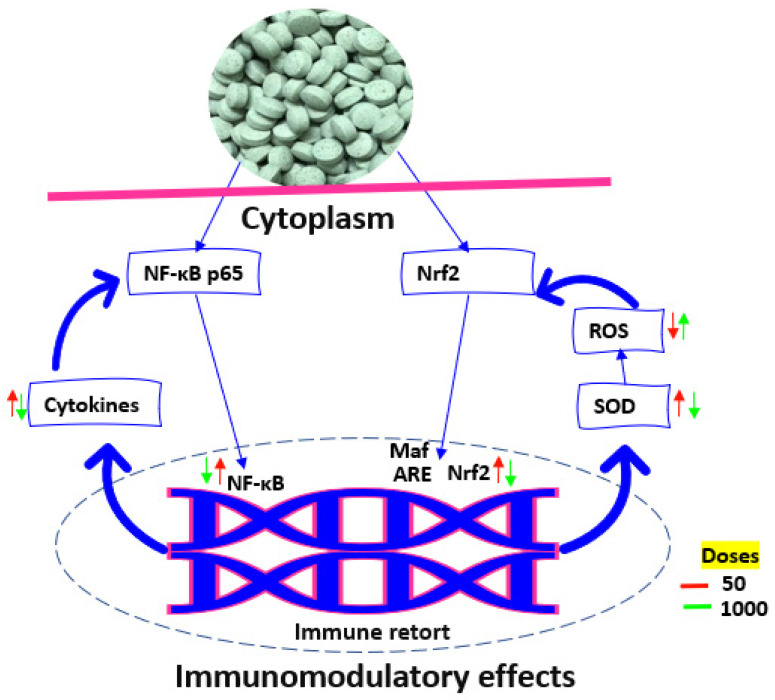
Representative diagram of the potential immunomodulatory effects of date seed pills on the RAW264.7 macrophages facilitated by impacts on nuclear NF-*κ*B and Nrf2 translocation.

**Table 1 foods-12-00784-t001:** Concentrations (mean ± SD) of the key flavonoids and phenolic substances found in the date seed extract before and after depolymerization.

Groups	Components	Quantity (g kg^−1^)
**Phenolic acids**	Protocatechuic acid	0.082 ± 0.003
**Hydroxycinnamic acids**	Caffeoyl shikimic acid	0.06 ± 0.001
**Flavan-3-ols**	Proanthocyanidin dimer B	0.44 ± 0.02
	Proanthocyanidin trimer 1	0.58 ± 0.01
	Catechin	0.3 ± 0.02
	Proanthocyanidin dimer B2	0.13 ± 0.01
	Proanthocyanidin trimer 2	0.04 ± 0.002
	Epicatechin	0.19 ± 0.03
**Flavones and flavonols**	Apigenin derivative	0.005 ± 0.001
	Quercetin derivative	0.04 ± 0.003
**Proanthocyanidins**	Epicatechin	45.9 ± 1.1
	Catechin	3.4 ± 0.4
**Total**		51.168

## Data Availability

Data will be accessible upon demand.

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
