# Peer review of "Immunomodulatory Effects of Spherical Date Seed Pills Industrially Fabricated on RAW264.7 Cells"

_foods, 2023, doi:10.3390/foods12040784_

Round 1
Reviewer 1 Report
It is necessary to carefully review the grammar of the document
It is necessary to describe in the materials and methods the different concentrations of maltodextrin, and gum arabic used to make the tablets.
Polyphenols, phenolic acids, flavonoids, or indistinct phenolic substances are used. Aren't flavonoids phenolic substances?
Phenolic acids are polyphenols; it is necessary to be clear about this
It is not clear to me what the x-axis of the figures represents. Is it the amount of tablet or the concentration of phenolic compounds that is in that amount of tablet? It would be worth clarifying... although there is a description, it is not entirely clear to me.
Author Response
|
It is necessary to carefully review the grammar of the document.
|
> Thank you for your suggestions, which indeed helped us to improve the quality of our manuscript. We have carefully revised the manuscript. Clearly, you will find these points by Track Changes. And we hope our revised and corrected manuscript satisfies you and gains your recommendation for publication. > We also wholly checked the text and fixed all the language issues. |
|
It is necessary to describe in the materials and methods the different concentrations of maltodextrin, and gum arabic used to make the tablets. |
> Revised as requested, L103-105. |
|
Polyphenols, phenolic acids, flavonoids, or indistinct phenolic substances are used. Aren't flavonoids phenolic substances? |
> Revised as requested. |
|
Phenolic acids are polyphenols; it is necessary to be clear about this |
> Revised as requested, yes, we totally agreed with you, polyphenols are the big group included phenolic acids, flavonoids, and so on. |
|
It is not clear to me what the x-axis of the figures represents. Is it the amount of tablet or the concentration of phenolic compounds that is in that amount of tablet? It would be worth clarifying... although there is a description, it is not entirely clear to me. |
> Revised as requested, this is the amount of each tablet, moreover, the figure legends were clarified in more detail. |

Reviewer 2 Report
The research "Immunomodulation effects of spherical date seed pills industrially fabricated on RAW264.7 cells" is impressive. Nonetheless, there are some comments I have attached that the authors need to incorporate in the manuscript to improve it.

Author Response
|
The research "Immunomodulation effects of spherical date seed pills industrially fabricated on RAW264.7 cells" is impressive. Nonetheless, there are some comments I have attached that the authors need to incorporate in the manuscript to improve it. |
> Thank you for your suggestions, which indeed helped us to improve the quality of our manuscript. We have carefully revised the manuscript. Clearly, you will find these points by Track Changes. And we hope our revised and corrected manuscript satisfies you and gains your recommendation for publication. > We have carefully fixed all the eleven comments you issued in the attached PDF which were highlighted in the main text. |

Round 2
Reviewer 1 Report
The authors have made the requested changes
Reviewer 2 Report
In my view, the authors have significantly improved their manuscript.